# Pleiotropic Actions of Aldehyde Reductase (AKR1A)

**DOI:** 10.3390/metabo11060343

**Published:** 2021-05-26

**Authors:** Junichi Fujii, Takujiro Homma, Satoshi Miyata, Motoko Takahashi

**Affiliations:** 1Department of Biochemistry and Molecular Biology, Graduate School of Medical Science, Yamagata University, Yamagata 990-9585, Japan; tkhomma@med.id.yamagata-u.ac.jp; 2Miyata Diabetes and Metabolism Clinic, 5-17-21 Fukushima, Fukushima-ku, Osaka 553-0003, Japan; miyata@miyata-dm.com; 3Department of Biochemistry, Sapporo Medical University School of Medicine, Sapporo 060-8556, Japan; takam@sapmed.ac.jp

**Keywords:** *Akr1a*, *Akr1b*, reductive detoxification, glycation, ascorbate synthesis, S-nitrosylation

## Abstract

We provide an overview of the physiological roles of aldehyde reductase (AKR1A) and also discuss the functions of aldose reductase (AKR1B) and other family members when necessary. Many types of aldehyde compounds are cytotoxic and some are even carcinogenic. Such toxic aldehydes are detoxified via the action of AKR in an NADPH-dependent manner and the resulting products may exert anti-diabetic and anti-tumorigenic activity. AKR1A is capable of reducing 3-deoxyglucosone and methylglyoxal, which are reactive intermediates that are involved in glycation, a non-enzymatic glycosylation reaction. Accordingly, AKR1A is thought to suppress the formation of advanced glycation end products (AGEs) and prevent diabetic complications. AKR1A and, in part, AKR1B are responsible for the conversion of d-glucuronate to l-gulonate which constitutes a process for ascorbate (vitamin C) synthesis in competent animals. AKR1A is also involved in the reduction of *S*-nitrosylated glutathione and coenzyme A and thereby suppresses the protein *S*-nitrosylation that occurs under conditions in which the production of nitric oxide is stimulated. As the physiological functions of AKR1A are currently not completely understood, the genetic modification of *Akr1a* could reveal the latent functions of AKR1A and differentiate it from other family members.

## 1. Introduction

Carbonyl compounds are produced by either enzymatic reactions during a variety of metabolic processes or by non-enzymatic reactions. Their production is particularly stimulated under conditions of oxidative stress caused by the production of excessive amounts of reactive oxygen species (ROS), which can be hazardous to our health. Enzymes in the aldo-keto reductase (AKR) superfamily catalyze the reduction of aldehyde groups to their corresponding alcohols in an NADPH-dependent manner [1]. As the AKR family of enzymes exhibit broad and overlapping substrate specificity, the definition of enzymes in reports, notably those published prior to gene identification, may be inconsistent.

Two closely related enzymes, aldehyde reductase (EC 1.1.1.2; AKR1A) and aldose reductase (EC 1.1.1.21; AKR1B), have been investigated from the standpoints of detoxification and the metabolism of bioactive compounds, notably in diabetes, cancer, and oxidative stress. AKR1A is abundantly present in the liver and kidney in a constitutive manner, while AKR1B is induced in some situations, such as hyperglycemia [2]. Although in many cases, AKR1A and AKR1B share common substrates, they also have some unique properties that differentiate them from one another. For instance, AKR1A catalyzes the denitrosylation of *S*-nitrosylated glutathione and coenzyme A [3,4], but, as of this writing, there are no reports of such activity for AKR1B. Conversely, while AKR1B participates in the reduction of glucose to sorbitol in the polyol pathway which is a cause for diabetic complications, glucose is not good substrate for AKR1A [5]. 

In this review article, we briefly overview the genetic and enzymatic properties of AKR1A and then discuss some diseases in which aldehyde metabolites play roles. The genetic name for aldehyde reductase is *Akr1a*, and, to specify species, an additional digit follows *Akr1a*, e.g., *Akr1a1* and *Akr1a4* for human and mouse aldehyde reductase, respectively [6,7]. As the genetic names for most genes do not define species, these names are sometimes misused in the literature. Here, we use AKR1A and *Akr1a* for the aldehyde reductase protein and the gene, respectively, without specifying species throughout this article, unless specification is essential.

## 2. Characteristics of the *Akr1a* Gene and the Regulation of Its Expression in Mammals

The properties of the AKR family have been extensively characterized from the standpoints of biochemistry and pathophysiology in the last half century [8]. However, the definition of enzymes in reports that appeared before identification of the genes for the AKRs and the establishment of the currently used nomenclature may be vague. There appears to be some confusion regarding the relationship between AKR1A and the other AKR family members. Updated information on the *Akr* genes is now available from the Ensembl data base [9] and proteins from the AKR super family website [7]. 

The complementary DNA for *Akr1a* was first cloned for humans [10] followed by other species, including the rat [11], pig [12], and mouse [13]. *Akr1b* cDNAs from these species were also cloned at about the same time, but the cloning of genomic DNA for *Akr1b* preceded that for *Akr1a*. The human *Akr1b* gene, which maps to chromosome region 7q35 [14], spans approximately 17 kb and consists of 10 exons [15], while a few pseudogenes are also present [16]. The human *Akr1a* gene (*Akr1a1*) was cloned somewhat later [8,17] and was found to be localized at chromosome region 1p32–p33 [17]. 

While the coding sequences of human and mouse *Akr1a* are divided into eight exons, the 5′-untranslated region in the mRNA was found to be encoded by alternative exons in the human gene (*Akr1a1*) [8]. Here, we adopt the numbering of the exons based on a report by Barski et al. [8]. *Akr1a1* is transcribed from alternative exon 1a and exon 1b, which are separated by a 1.3-kb intron (Figure 1). A large 9.2-kb intron that contains the translational initiation codon ATG precedes exon 2. Accordingly, the two splicing mRNA variants carrying different 5′-untranslated regions are transcribed from *Akr1a1*, but the translation of these variants produces identical AKR1A protein products. These splicing variants of mRNA are expressed in all tissues to similar extents [8], and hence the physiological significance of the alternative splicing of *Akr1a1* has not been clearly understood until now. After exon 2, a large intron is present again, and the following exon 3 to exon 9 exist within narrow DNA sequences. No alternative exon has been reported for the mouse *Akr1a4* gene, which is a slightly smaller than *Akr1a1* and contains slightly larger introns among the exons in the 3′-region. 

There is a consensus sequence for binding of the transcription factors C/EBP-homologous protein (CHOP) upstream from the transcription initiation site of *Akr1a1* [8], which is a transcriptional regulatory factor induced by transcriptional factor ATF4 upon endoplasmic reticulum stress and is involved in the cell death pathway [18]. Mutation in the consensus binding sequences for CHOP abolishes the promoter activity of *Akr1a1* suggesting that CHOP is a major contributor to the regulation of the gene expression. Transcription factor NF-E2-related factor 2 (NRF2) is the master transcriptional regulatory factor for the expression of multiples genes that are involved in the antioxidation and detoxification of xenobiotic compounds [19]. While *Akr1c1* is highly inducible by activated NRF2, *Akr1a1* and some other *AKRs* are also substantially increased via the knockdown of the Kelch-like ECH-associated protein 1 (Keap1), a suppressor of NRF2 [20]. These observations suggest that *Akr1a1* is involved in the amelioration of such stress conditions.

## 3. Protein Structure and Catalytic Reaction of AKR1A

AKR1A is an approximately 38 kDa monomeric protein and is widely distributed in tissues, but is especially abundant in the kidney and liver [2]. Replacing the lysine residue at position 262 in AKR1A with a methionine residue results in a 97-fold increase in the *K_m_* for NADPH and an 86-fold increase for d,l-glyceraldehyde, suggesting that lysine-262 has a significant role in NADPH binding as well as substrate binding [21]. Mutant AKR1B in the corresponding lysine also shows a similar increase in *K_m_*. The three-dimensional structure of the AKR1A protein has been determined based on crystallography data [12]. The first preliminary data on the crystallography was reported for human AKR1A (AKR1A1) and porcine AKR1A (AKR1A2) [22], and the detailed structure of AKR1A2 was subsequently reported [23]. The alignment of the amino acids of AKR family proteins suggest that the (α, β)_8_-barrel fold forms a scaffold for the NAD(P)H-dependent catalytic activity of these proteins [1]. The active site is located at the carboxy terminus of the (α, β)_8_ barrel in both AKR1A2 and AKR1B2. However, the *N*^ε2^ of the imidazole ring of histidine-113 in AKR1A2 interacts with the amide group of the nicotinamide ring of NADPH, which is different from AKR1B2 [22] and may cause a difference in substrate specificity. The hydroxy group of tyrosine-50 in AKR1A2 appears to form hydrogen bonds with the oxygen moiety of the aldehyde group of the substrate [24]. The reduction of the aldehyde substrate appears to follow a Bi–Bi ordered catalytic mechanism in which NADPH binds first and, after oxidation, leaves last, as has been found in other AKR family enzymes [25]. The binding of NADPH induces a conformation change in the enzyme, which facilitates substrate binding. The tertiary complex of AKR1A2 with NADPH and 3,5-dichlorosalicylic acid, a potent inhibitor for AKR1C1, reveals the structural basis of the difference in their inhibitory efficiency [26]. The subsequent crystallization of mouse AKR1A (AKR1A4) and binding-affinity measurements actually indicate a very high affinity binding of NADPH to the enzyme (*K_d_* = 144.6 nM) [27].

The regulatory mechanism of the AKR1A activity in response to stimuli is largely unknown, with the exception of that by SMAR1. SMAR1 is a tumor suppressor that is involved in cell cycle regulation and reportedly inhibits mouse AKR1A (AKR1A4) via a direct interaction [28]. Activation of ataxia telangiectasia mutated (ATM) kinase leads to the dissociation of the SMAR1–AKR1A4 complex, which results in the activation of AKR1A4 and, at the same time, cell cycle arrest via the translocation of SMAR1 to the nucleus. An elevation in AKR1A4 activity caused by the downregulation of SMAR1 leads to the protection of malignant cells against toxic aldehydes which are produced during activated metabolism in tumor cells and treatment with anti-cancer agents. Nevertheless, the co-crystallization of AKR1A4 and a SMAR1-derived 44-amino acid peptide has been unsuccessful, which suggests that another molecule is required for their interaction [27].

## 4. NADPH-Dependent Reduction of a Variety of Aldehyde Compounds

Redox homeostasis is maintained by a unique enzymatic function attributable to each group of oxidoreductase enzymes [29], and NADPH-dependent aldehyde reduction is the function of AKR family proteins. A variety of compounds that include lipid- and carbohydrate-derived aldehydes and metabolic intermediates can be substrates for these enzymes [30]. Typical examples of reactions catalyzed by AKR1A or overlapping reactions by other AKR family members are addressed below. 

### 4.1. Aldehydes Produced by Lipid Peroxidation

O’Connor et al. [31] prepared recombinant AKR1A as well as several other AKR1 family proteins and examined their substrate specificities in detail. As a result, a broad spectrum of aldehyde compounds were found to be substrates for AKR1A. Many of these aldehyde compounds are also substrates for AKR1B, but the catalytic efficiency of AKR1A for aromatic and aliphatic aldehydes is higher than that for AKR1B. Oxidation preferentially occurs in unsaturated fatty acids with the production of lipid peroxides [32]. The resulting lipid peroxidation products then undergo cleavage and release smaller aldehyde compounds, among which 4-hydroxy 2-nonenal (HNE) and malondialdehyde are abundant components [33]. Notably, HNE is a highly toxic aldehyde that is involved in the pathogenesis of a variety of diseases including inflammatory and neurodegenerative diseases [34]. HNE is in fact a substrate for enzymes of the AKR family, which include AKR1A and, importantly, AKR1C, and is reduced to the less toxic 1,4-dihydroxy-2-nonene [35] (Figure 2A).

Acrolein (2-propenal), which is produced in lesser abundance by lipid peroxidation reactions compared to HNE but also arises largely from tobacco smoke, heated cooking oil, and air pollutants. Moreover, acrolein is a metabolite of cyclophosphamide, a popular alkylating agent with anti-cancer effects, and appears to cause side effects in several organs [36]. The cytotoxicity of acrolein is quite high among the lipid-derived aldehydes [37], and the formation of a DNA adduct with acrolein leads to mutations in genes and hence consequent tumorigenesis [38]. Thus, the detoxification of acrolein is important in preventing related diseases, notably chronic obstructive pulmonary disease (COPD) [39]. While acrolein adducts of proteins are numerous in the male genital tract and testes, AKR1A and AKR1B are also abundantly present there, suggesting the detoxification of acrolein by these enzymes [40]. The AKR-catalyzed reduction of acrolein results in the production of allyl alcohol [41] (Figure 2B). AKR1A-deficient mouse embryonic fibroblasts become more sensitive to the cytotoxic action of acrolein [42]. The overexpression of human *Akr1a* makes the cells more resistant to acrolein cytotoxicity. It has been reported that some AKR family members, namely AKR1B, AKR1B7, AKR1C, AKR7A1, and AKR7A2, are also involved in the detoxification of acrolein [43,44,45,46,47,48].

### 4.2. Methylglyoxal as a Carbohydrate-Derived Substrate

Methylglyoxal, a dicarbonyl compound, is generated non-enzymatically from intermediates in the glycolysis pathway, notably from glyceraldehyde 3-phosphate, and also peroxidase-catalyzed reactions. Methylglyoxal is cytotoxic and may cause apoptosis at relatively high doses [49], although methylglyoxal is much less cytotoxic than acrolein [42]. AKR, including AKR1A and AKR1B, reduces methylglyoxal and results in the formation of either acetol (95%) or lactoaldehye (5%) [50] (Figure 2C). Glyoxalase-I and –II are enzymes that metabolize methylglyoxal to unreactive d-lactate [51]. The overexpression of glyoxalase-I completely prevents the formation of advanced glycation end products (AGEs) in cells, indicating that methylglyoxal is a precursor for AGEs [52]. Given the presence of glyoxalases, further studies using genetically modified animals would be required to evaluate the physiological role of the AKR1A-catalyzed detoxification of methylglyoxal in an in vivo situation. 

### 4.3. Aldehydes Produced from Monoamine Metabolism

Monoamine oxidase catalyzes the conversion of neurotransmitter amines to aldehyde forms, which are then reduced to alcohols by NADPH-dependent reductases or NAD^+^-dependent dehydrogenases [53]. For example, the neurotransmitter γ-aminobutyric acid (GABA) is converted into succinic semialdehyde by the action of monoamine oxidase or GABA aminotransferase and is then oxidized to succinic acid by succinic semialdehyde dehydrogenase or reduced to γ-hydroxybutyrate by AKRs [29,54,55] (Figure 2D). While current information suggests that AKR1A is involved in the production of γ-hydroxybutyrate, siRNA-mediated AKR1A knockdown in a human astrocytoma cell line (1321N1) has little effect, suggesting the compensatory action of other enzymes such as AKR7A2 [56]. The following study using HepG2 confirmed that AKR1A does not play a role in the synthesis of γ-hydroxybutyrate in cells [57]. These observations suggest that AKR1A may not have a major role in this reaction. 

### 4.4. Roles of Akr1a in Drug Metabolism

A literature survey of oxidoreductase enzymes (cytochrome P450, microsomal flavin-containing monooxygenase, monoamine oxidase, and AKR) regarding drug metabolisms suggests that P450s participate in more than 90% of the reactions in the human body [58]. Nevertheless, in some drugs, an aldehyde group is essential for exerting medicinal effects or the accompanying side effects and hence the reduction of the group neutralizes their effects. The pharmacokinetics of anthracycline anti-cancer agents, such as doxorubicin and daunorubicin, vary considerably in individual patients [59] and appear to be associated with the protein levels of some enzymes, including AKR1A, AKR1B, and carbonyl reductase (CBR) 1 [60,61]. In fact, the association of competency in the conversion of doxorubicin to doxorubicinol and interactions with and the activities of AKR1 and CBR proteins has been reported [62,63,64] (Figure 2E). It has also been suggested that AKR1C2 is involved in the reductive inactivation of anthracyclines, but an enzyme assay using a recombinant protein confirmed a decrease in the drug effects by AKR1A1, but not by AKR1C2 [65]. 

Some of single nucleotide polymorphisms in *Akr1a1* cause amino acid replacements in AKR1A1 and may be associated with sensitivity to anthracycline. When recombinant AKR1A1 proteins with amino acid the replacements, glutamate-55 to aspartate and asparagine-52 to serine, were produced and subjected to enzymatic analyses, both mutant AKR1A1 proteins showed a reduced affinity for daunorubicin [66]. Thus, these allelic AKR1A1 variants may be associated with variations in the cardiotoxic side effects of doxorubicin and daunorubicin. Among the drugs that are administered in combination with daunorubicin for therapeutic purpose, all-trans-retinoic acid effectively inhibits AKRs and thereby sustains therapeutic efficacy [67]. AKR1C3 is the most strongly inhibited by all-*trans*-retinoic acid, but inhibitory action on AKR1A1 is limited. The anti-inflammatory drug tiaprofenic acid is also reduced by AKR1A1 and other AKRs, although the activity is much less than CBR1 [68].

### 4.5. Roles of AKR1A in the Synthesis of Bioactive Compounds

The roles of AKR in steroid metabolism have also been implied. AKR1A preferentially catalyzes the reduction of aldehydes derived from 17-deoxycorticosteroids, while AKR1B reduces aldehydes derived from cortisol and corticosterone with a similar efficacy [69]. In later studies, however, AKR1C members were found to largely contribute to the reductive conversion of 3-ketosteroid, 17-ketosteroid, and 20-ketosteroid, while AKR1D1 acts on 5β-steroids [70]. Hence, the inhibition of AKR1C3 appears to be a potential therapeutic approach for the treatment of both hormone-dependent and hormone-independent cancers [71]. 

Prostaglandins (PGs) are short-lived lipid mediators that are produced from arachidonate. PGF_2__α_ is widely distributed in mammalian organs and exhibits a variety of biological functions. Several enzymes in the AKR family have been reportedly involved in the synthesis of prostaglandin F2 [72]. Whereas AKR1C appears to be robustly involved in the prostaglandin F2 synthesis, subsequent studies have also revealed that AKR1B has a role in the production of PGF_2__α_ [73,74]. It has also been reported that both AKR1A and AKR1B are potent prostaglandin F2 synthases among other AKR family members [75]. A systematic study on the related compounds will be required to reveal whole picture regarding the individual role of AKR family members in PG metabolic pathways. 

## 5. AKR1A as a Suppressor of Diabetic Complications

Multiple metabolic disorders have been proposed as causes of diabetic complications [76]. In the polyol pathway that is stimulated under conditions of hyperglycemia, AKR1B catalyzes the oxidative conversion of glucose to sorbitol, which is then reduced to fructose by the action of sorbitol dehydrogenase (SDH) (Figure 3). The accumulation of sorbitol increases osmotic pressure in cells, leading to osmotic dysfunction in the eye and kidney [77]. The glycation reaction is a non-enzymatic reaction that involves reactions reducing carbohydrates, as represented by glucose, and an amino group in amino acids, proteins, and other compounds. In fact, elevated levels of glycated hemoglobin (hemoglobin A1c; HbA1c) is the hallmark for hyperglycemia in diabetic patients. As glucose is the most abundant free reducing sugar, glycation advances at an accelerated rate under hyperglycemia and is regarded to be one of the causes for diabetic complications [78]. 

### 5.1. Glycation as a Potent Non-Enzymatic Reaction Responsible for Diabetic Complications

The reaction of glucose with an amino group initiates the glycation reaction, which becomes an Amadori adduct of the protein in the intermediate process. 3-Deoxyglucosone (3-DG) is a reactive dicarbonyl compound that is formed in vivo from Amadori adducts of proteins [79]. The resulting 3-DG further reacts with proteins, which results in the production of AGEs and causes diabetic complications [80]. The resulting AGEs cause protein cross-linking and directly contribute to multiple organ failure, including the cardiovascular system, the central nervous system, and the kidneys [81,82]. AGEs also exert detrimental effects via receptors for AGEs, RAGEs, by stimulating the production of reactive oxygen species and inflammation. In addition to glucose, fructose is in fact a more potent glycating agent and stimulates glycation. As the polyol pathway is very active in the lens, the levels of fructose-conjugated crystallins are elevated in rat lenses under conditions of hyperglycemia, as detected by an anti-fructosylated lysine antibody [83,84]. Fructose also can be phosphorylated to fructose 3-phosphate, which then decomposes to 3-DG [85]. Moreover, because fructose 3-phosphate *per se* stimulates the glycation reaction, the activation of the polyol pathway enhances the glycation reaction. Accordingly, glycation reactions are especially active in lens proteins, which have long half-life and induce the formation of cataracts in diabetic patients. 

### 5.2. AKR1A Is an Enzyme Detoxifying 3-DG

Dicarbonyl compounds, which are highly reactive intermediates in the glycation reaction, have been postulated to contribute to the development of diabetic complications both directly and via the formation of AGEs [86,87,88,89,90,91,92]. The body possesses multiple protective machineries that function to suppress glycation. 3-DG is reductively converted into 3-deoxyfructose (3-DF), which tends to suppress the glycation reaction [93]. The levels of 3-DG and 3-DF have been reported to be increased in human plasma and urine in diabetes patients [94,95]. An enzyme that catalyzes the reduction of 3-DG to 3-DF was purified for the first time from porcine liver and is designated as an NADPH-dependent 2-oxoaldehyde reductase [96]. Kanazu et al. [97] isolated 3-DG reducing enzymes from the rats, pigs, and humans and reported on their enzymological and immunological properties, which support the identity of the 3-DG reducing enzyme to AKR1A. The catalytic action of AKR1A and AKR1B on carbohydrate 2-oxoaldehydes (osones), including 3-DG, has been characterized enzymatically [98]. Thus, it has been suggested that AKR1A plays a role as a major protective mechanism against carbonyl stress by catalyzing the reduction of dicarbonyl compounds such as 3-DG to inert compounds. 

### 5.3. Protection of Akr1a against Diabetic Complications

We reported on the isolation of the dominant 3-DG-reducing enzyme from rat liver and determined the amino acids sequences of nine peptides after lysyl-endopeptidase digestion, which resulted in the identification of AKIR1A as the 3-DG-reducing enzyme [11]. After the cloning of rat *Akr1a* cDNA, the transient expression in COS-1 cells followed by enzymatic assay confirmed the identity. At about the same time, it was reported that both kidney AKR1A and AKR1B reduce 3-DG [99,100]. Enzyme kinetic data imply that AKR1A catalyzes the NADPH-dependent reduction of not only 3-DG but also methylglyoxal. As methylglyoxal is also a cytotoxic 2-oxoaldehyde compound and is involved information of AGEs under hyperglycemia [80], the AKR1A-catalyzed reduction of 3-DG and methylglyoxal would, in any case, prevent AGE formation and hence suppress diabetic complications. On the other hand, AKR1A itself undergoes glycation and inactivation [101], suggesting that a decreased AKR1A activity may accelerate diabetic complications and carcinogenesis in diabetic patients.

We have established and used *Akr1a*-knockout (KO) mice to clarify the etiological contribution of 3-DG and the protective role of AKR1A in the development of diabetic nephropathy in vivo. Based on our results [102], plasma 3-DG levels were significantly increased in wild-type (WT) diabetic mice compared with nondiabetic WT mice at 6 months after a streptozotocin injection. This phenomenon was notably accelerated in *Akr1a*-KO mice. Interestingly, plasma 3-DG levels in nondiabetic *Akr1a*-KO mice were elevated to a level comparable to that of diabetic WT mice (Figure 4(Aa)). It was also revealed that plasma 3-DG levels in *Akr1a*-KO mice were higher than that of WT mice with comparable levels of HbA1c (Figure 4(Ab)). In proportion to plasma 3-DG levels, an increase in the formation of pyrraline, one of the 3-DG-derived AGEs, was observed in the renal glomeruli, so that the tissues of diabetic *Akr1a*-KO mice showed a prominent accumulation of pyrraline (Figure 4B). The periodic acid–methenamine (PAM) staining of renal tissues showed an accelerated expansion of the mesangial matrix, a characteristic pathological change associated with diabetic nephropathy, in the glomeruli of diabetic *Akr1a*-KO mice (Figure 4C). The earliest manifestation of a renal disorder in diabetes is an increase in the rate of urinary excretion of albumin (ACR). The ACR gradually increased with time after the induction of diabetes, and was accelerated in the case of *Akr1a*-KO mice (Figure 4D). Interestingly, *Akr1a*-KO mice showed the same phenotype of diabetic nephropathy even under non-diabetic conditions with a moderate increase in plasma 3-DG level. These findings indicate that an increased formation of 3-DG plays an important role in hyperglycemia-related mechanisms by which diabetic nephropathy is developed, and that AKR1A exerts a protective function against diabetic complications. 

## 6. Roles of AKR1A in Ascorbate Biosynthesis and Action in Mice 

In initial studies, AKR1A was found to reduce d-glucuronate to l-gulonate [103,104], which is now considered to be a part of metabolic pathway for the synthesis of ascorbate in competent animals (Figure 5) [105]. The ascorbate synthesis pathway is defective in primates and some other species that lack the *Gulo* encoding the rate-determining enzyme responsible for the final step of ascorbate synthesis [106]. Thus, primary functions of AKR1A and the rest of the enzymes for the ascorbate-synthesizing process are attributable to metabolic processes other than for ascorbate synthesis in such animals that are unable to synthesize ascorbate. 

Ascorbate is a required micronutrient in incompetent animals and designated as vitamin C in primates. Due to its electron-donating ability, ascorbate acts as a major antioxidant by suppressing the toxic effects of lipid peroxides via the reductive recycling of tocopherol (vitamin E) radicals [107]. Ascorbate is also a cofactor for α-ketoglutarate-dependent non-heme iron containing dioxygenases that catalyze prolyl hydroxylation, and the demethylation of proteins and DNA [108]. As most animals, as represented by rodents, produce ascorbate, they are not suitable as human models in investigations of the physiological function of ascorbate. 

By employing genetically deficient mice, Gabbay et al. [109] reported, for the first time, that the catalytic reduction of d-glucuronate to l-gulonate by AKR1A and partly by AKR1B is actually involved in the synthesis of ascorbate in vivo. In mice, the contribution of AKR1A and AKR1B is approximately 85% and 15%, respectively. Accordingly, while *Akr1a*- KO mice consequently show severe osteopenia and spontaneous fractures due to an ascorbate deficiency, *Akr1b*- KO mice do not. As the liver is the central organ where ascorbate is synthesized, the dominant presence of AKR1A compared to AKR1B appears to account for the difference in their contribution. Mice in which eGFP is knocked-in on the *Akr1a4* locus have been established and may be useful model animals for investigating osteoporosis caused by *Akr1a4* ablation [110]. 

We have also demonstrated that AKR1A catalyzes the conversion of d-glucurono-**γ**-lactone to l-gulono-**γ**-lactone, which constitutes the alternative pathway for the conversion of d-glucuronate to l-gulono-**γ**-lactone [111]. Since *Akr1a*-KO mice have subsequently been further employed in some studies related to ascorbate, we attempted to clarify the physiological functions of AKR1A itself as well as ascorbate under in vivo conditions. While *Akr1a*-KO mice show a prolonged anesthetic reaction to pentobarbital, ascorbate reverses it, which implies that ascorbate has a role in the central nervous system recovering from anesthesia [112]. The application of the *Akr1a*-KO mice in a water maze test showed that an ascorbate insufficiency impaired spatial memory formation in juvenile mice, but not in adult *Akr1a*-KO mice, without showing any noticeable hippocampal damage [113]. Considering the role of the brain in the long-term retention of ascorbate, these results suggest that ascorbate plays important roles in neuronal actions, especially in juvenile mice. Developmental divergence is seen in *Akr1a*-null neonatal mice, but not in *Akr1a*-heteozygous neonatal mice that are delivered from an *Akr1a*-null mother [114]. An inverse correlation was observed between plasma ascorbate levels and corticosterone levels in the *Akr1a*-KO neonates. As the ability to synthesize ascorbate is low in the fetus, maternally derived ascorbate may not be evenly transferred to embryos in polytocous mice, thus causing corticosterone levels and consequent embryonic development to be altered.

Another example of ascorbate function is protection against toxicants. Hepatic steatosis induced by carbon tetrachloride is more severe in the livers of *Akr1a*-KO mice compared to WT mice and is restored to the levels in WT mice by ascorbate supplementation [115]. While an overdose of acetaminophen (APAP) causes more severe hepatic injury in *Akr1a*-KO mice, ascorbate supplementation effectively rescues this fatal phenotype [116,117]. These improvements, at least partly, appear to be associated with the antioxidative function of ascorbate as this relates to the elimination of superoxide radicals. The ablation of both *Akr1a* and superoxide dismutase 1 (*Sod1*) causes premature death in mice after about two weeks, but the life span of the double deficient mice is markedly prolonged by ascorbate supplementation [118]. Accordingly, it is conceivable that the health of *Sod1*-KO mice is largely maintained by the endogenous synthesis of ascorbate. These collective data suggest that the intake of sufficient ascorbate is of prime importance for ascorbate-incompetent animals, including humans. The reason for why ascorbate-incompetent animals, including primates, can be prosperous, despite the inability to synthesize such an important metabolite, is an open question. 

## 7. *Akr1a* Functions That Are Not Directly Associated with Diabetic Complications or Ascorbate Synthesis

Pathological phenotypes observed in the *Akr1a*-KO mice are not always rescued by ascorbate supplementation, raising the question as to whether such phenotypes are caused by a defect other than glucuronate reduction by AKR1A and indicating the usefulness of genetically modified mice to unveil the novel functions of AKR1A. Ascorbate is the established electron donor for collagen synthesis and, consequently, a severe insufficiency causes scurvy in humans. However, collagen synthesis and kidney fibrosis, which is induced by unilateral ureteral obstruction, are not affected by an ascorbate deficiency (~10% of WT mice) in *Akr1a*-KO mice [119]. The results suggest the existence of other compounds that compensate for ascorbate in vivo. Thus, the *Akr1a*-KO mice could be useful model animals to determine if ascorbate is actually involved in the target reaction or whether other compounds may compensate for its absence. 

The use of the resident–intruder test revealed that the *Akr1a*-KO mice exhibit more aggressive phenotypes than WT control mice, and this behavior cannot be rescued by ascorbate supplementation [120]. Thus, the heightened aggression in the *Akr1a*-KO mice is independent of ascorbate synthesis. Interestingly, the plasma levels of corticosterone are increased, which implies the existence of an association between *Akr1a* ablation and altered steroid metabolism as described above. *Gulo*-KO mice whose drinking water was supplemented with ascorbate (220 ppm), necessary for generating a chronic low-ascorbate status in the brain, showed a behavioral abnormality [121]. The mice moved less actively, suggesting an association of the neuronal function in the central nervous system with ascorbate, although no evidence of cognitive, anxiety, or sensorimotor-gating problems has been reported. On the other hand, when *Akr1a*-KO mice are subjected to a treadmill test, running performance is higher in the *Akr1a*-KO mice than the WT mice irrespective of ascorbate status [122]. Free fatty acid levels tend to be initially high in blood plasma, and, after the exercise period, the levels of blood glucose and liver glycogen are preserved in the *Akr1a*-KO mice. Thus, preserving blood glucose and the preferential utilization of fatty acids may be combined to contribute to the improved athletic performance. PGC-1α, a key regulator of energy metabolism [123], and the products of downstream target genes are constitutively at high levels in the *Akr1a*-KO mice. PGC-1α is the coactivator for the glucocorticoid receptor and peroxisome proliferator-activated receptors (PPARs), which are activated by a variety of lipid peroxidation products [124]. While corticosterone levels are elevated in the blood [120], lipid-derived aldehydes may also increase due to an AKR1A deficiency and activate PPARs in the *Akr1a*-KO mice. Thus, activated glucocorticoid receptors and/or PPARs may form a complex with PGC-1α and be involved in the enhanced exercise ability of the *Akr1a*-KO mice. 

Thioacetamide (TAA) is a hepatotoxicant that undergoes bioactivation via the cytochrome P450 (CYP). *Cyp2e1*-KO mice do not develop an apparent liver injury after treatment with TAA, suggesting that CYP2E1 is the major enzyme responsible for TAA bioactivation [44]. As the treatment of the *Akr1a*-KO mice with TAA also showed resistance against its hepatotoxicity compared to WT mice [125], the results suggest a possible role of AKR1A in the bioactivation of TAA similar to that for CYP2E1. However, TAA does not contain an aldehyde group, suggesting that the action of AKR1A on TAA is a currently unknown, indirect mechanism. 

Studies of drug-induced hepatic injuries in *Akr1a*-KO mice caused by an APAP overdose show results that cannot be explained by an ascorbate deficiency alone. *Akr1a*-KO mice are highly vulnerable to an overdose of APAP compared to the WT mice, and ascorbate supplementation renders the *Akr1a*-KO mouse to be markedly resistant to APAP hepatotoxicity [116]. This may, at first glance, imply that ascorbate exerts a strong protection against APAP hepatotoxicity. Under conditions of an iron load, the hepatotoxicity of APAP was comparable between WT and the *Akr1a*-KO mice, and ascorbate supplementation dramatically ameliorated the APAP-induced hepatic injury in the iron-loaded *Akr1a*-KO mice but had no effect on the iron-loaded WT mice [124]. The ascorbate contents were actually higher in the ascorbate-supplemented WT mice than in the ascorbate-supplemented *Akr1a*-KO mice due to intrinsic ascorbate synthesis. It therefore appears that ascorbate cannot be used to rationalize resistance of the iron-loaded mice. 

Our metabolomic study indicated that the *Akr1a*-KO mice had an extensive accumulation of d-glucuronate (41-fold in the *Akr1a*-KO vs. the WT mice) and the levels of oxidized metabolite saccharate (102-fold in the *Akr1a*-KO vs. the WT mice) in the liver [111]. As a result, it is conceivable that the restriction in carbohydrate flow to ascorbate synthesis enhanced the detoxification of APAP by stimulating glucuronate conjugation, leading to the *Akr1a*-KO mouse being more resistant to APAP under conditions of an iron overload [126]. This potential elevation in the glucuronidation reaction may be more advantageous to the ascorbate-incompetent animals compared to the ascorbate-competent animals, leading to them to prosper in the current animal kingdom. 

## 8. AKR1A Plays Antithetic Roles in Cancer Development

An elevated expression of AKR1A is observed in many cancer cells [127,128], and AKR1A functions are associated with cancer development in two conflicting ways. One is protection against carcinogenesis via the reductive detoxification of carcinogens or secondarily produced genotoxic aldehydes, and the other is the promotion of cancer development by disabling cancer treatment (Figure 6). Aldehyde compounds react with a variety of molecules, including proteins and nucleotide bases, and form adducts, which may transmit a signal or cause dysfunction in cells [129]. As the resulting responses to aldehyde modification depend on the target on which they act, AKR1A-mediated reduction would be expected to sometimes exert conflicting effects. When AKR1A reacts with mutagenic aldehydes in normal cells, the reaction leads to the suppression of carcinogenesis. The risk of non-Hodgkin lymphoma increases by 1.7-fold (*p* value = 0.0047) in individuals carrying a homozygous *Akr1a1* (282T -> C) variant [130]. Although how this single nucleotide polymorphism affects AKR1A1 activity is not known, the variant may be involved in carcinogenesis in cases of decreasing AKRA1 activity. On the other hand, the tumoricidal activities of anti-cancer drugs may be diminished by AKR1A-catalyzed reduction, as typically seen in doxorubicin, as mentioned above (Figure 2E). In radiation therapy, ionization radiation produces ROS, which then exhibits tumoricidal activities. Cytotoxic aldehydes that are produced by ROS-medicated oxidation are partly involved in the tumoricidal action [131]. In proteomic analyses, an elevated expression of AKR1A was observed in radiation-resistant laryngeal cancer-derived HEp-2 cells [132], and the AKR1A-involved suppression of p53 function appears to be responsible for the resistance to radiation [133]. 

The participation of AKRs in the process of chemical carcinogenesis has been extensively investigated. Benzene, a ubiquitous environmental pollutant arising from automotive emissions and cigarette smoke, undergoes metabolic conversion and can lead to the development of anemia and leukemia. *Trans*, *trans*-muconaldehyde, a cytotoxic metabolite of benzene, is mainly detoxified by alcohol dehydrogenase (ADH1) and partly by AKR1A [134]. Other examples are polycyclic aromatic hydrocarbons that are produced from tobacco smoke and other types of combustion and are metabolically activated into carcinogens. AKR1A1 preferentially oxidizes (−)-[3R,4R]-dihydroxy-3,4-dihydrobenz[a]anthracene [135] and proximate carcinogen *trans*-dihydrodiols to their corresponding *o*-quinones [136]. Benzo[a]pyrene, a well-known carcinogen, is metabolized to a proximate carcinogen, (+/−)-7,8-dihydroxy-7,8-dihydrobenzo[a]pyrene (BP-7,8-diol), which then experiences further metabolism and eventually becomes a fully carcinogenic compound. While AKR1A1 and AKR1C1–AKR1C4 catalyze the formation of benzo[a]pyrene-7,8-dione (BP-7,8-dione) [137], P450 1A1/P450 1B1 catalyzes the formation of (+/−)-*trans*-7,8-dihydroxy-9α,10α-epoxy-7,8,9,10-tetrahydrobenzo[a]pyrene [138]. Thus, AKR1A1 involves the bioactivation of benzo[a]pyrene depending on the situation. Further analyses of the role of AKR1A1 in the activation of benzo[a]pyrene-7,8-diol have revealed that AKR1A1 has bifunctional roles in carcinogenesis [139], i.e., AKR1A1 directly catalyzes the bioactivation of the compound and also indirectly stimulates the reaction via the induction of another metabolic enzyme, P4501B1, in human bronchoalveolar cells. Moreover, during the AKR-catalyzed oxidation of benzo[a]pyrene to BP-7,8-dione, ROS are produced and coordinately damage DNA in human lung A549 adenocarcinoma cells [140]. On the contrary, overexpressed *Akr1a* reportedly protects cells from the toxic effects of BP-7,8-diol in A549 cells [141], although cell proliferation or cell cycle progression is not affected irrespective of the presence of benzo[a]pyrene-7,8-dione in human bronchoalveolar carcinoma H358 cells [142]. 

Diethylnitrosamine (DEN)-induced liver carcinogenesis is another example of this issue. The expression of *Akr1a*, as well as glutathione S-transferase and alcohol dehydrogenase, is elevated during DEN-induced carcinogenesis in the rat liver [143]. As malondialdehyde and HNE induce the expression of *Akr1a* [144], lipid peroxidation caused by the DEN treatment may be responsible for the induction of *Akr1a*. The resulting AKR1A may play roles in the reductive detoxification of the aldehyde compounds and, hence, act as an anti-tumor agent. Consistent with the important role in anti-carcinogenic action, the treatment of *Akr1a*-KO mice with DEN results in the development of numerous nodules [145]. As ascorbate supplementation markedly reduces the numbers of nodules in the *Akr1a*-KO mice, elevated ascorbate levels that occur concomitantly with the increase in AKR1A may be partly involved in anti-carcinogenesis in the case of the rodent models. 

While levels of AKR1B have been reported to be elevated in rat primary hepatoma and human hepatoma cell lines, the levels of AKR1A were not [146]. The Long–Evans with a cinnamon color (LEC) rat is a Wilson’s disease model, which characteristically accumulates copper in the liver due to a mutation in the copper transporter gene *Atp7b*, and hereditarily develops hepatitis and hepatoma at an advanced stage and shows increased levels of AKR1B [147]. However, expression of AKR1A is again unchanged during the hepatocarcinogenic process again. Other research groups have reported that the expression of *Akr1a* and *Akr1b* is relatively constant in four HCC cell lines, whereas *Akr1c* expression appears to be associated with hepatocellular carcinoma [148]. Thus, the actions of AKR1A1, whether anti-cancer or carcinogenic, appear to depend on the types of cells and also on the timing of the carcinogenic process. 

## 9. AKR1A Catalyzes the Reduction of S-Nitrosoglutathione and S-Nitroso-Coenzyme A

Nitric oxide (NO) is produced through both enzymatic reactions and non-enzymatic reactions and reaches abundant levels, especially in the cardiovascular system and under inflammatory conditions [149]. While the sulfhydryl group undergoes oxidative modification by NO to form *S*-nitrosothiol (SNO), this process is also mediated by enzymes designated as SNO synthases [150]. *S*-nitrosylation, which occurs to an enhanced extent under conditions of an excessive production of NO, is involved in cellular signaling pathways and nitrosative stress [151,152]. Low molecular weight thiol compounds, notably cysteine and glutathione (GSH) in cells, are amenable to *S*-nitrosylation and are converted to *S*-nitroso-cysteine and *S*-nitroso-glutathione (GSNO), respectively, which act as trans-nitrosylation donors. GSNO is formed more abundantly than S-nitroso-cysteine due to the dominant presence of GSH in cells. GSNO reductases, comprising GSH/GSH-dependent formaldehyde dehydrogenase, class III alcohol dehydrogenase (ADH5), and thioredoxin/thioredoxin reductase act as denitrosylases [153]. The balance between *S*-nitroyslation and denitrosylation regulates the signaling action of SNO and avoids nitrosative stress [4,154]. 

While ADH5 acts as a major GSNO reductase in an NADH-dependent manner [155,156], GSNO is also reduced in an NADPH-dependent manner. CBR1 is an enzyme that functions as an NADPH-dependent GSNO reductase [157]. While the X-ray crystallographic structure of human CBR1 reveals a GSH binding site, enzymatic analyses imply that GSNO is the most preferential substrate (*K_m_* = 30 µM, *k_cat_* = 450 min^−1^) among the known carbonyl substrates. A recent study demonstrated that AKR1A1 also exerts NADPH-dependent GSNO reductase activity, which results in the production of glutathione-sulfinamide [3] (Figure 7). Coenzyme A (CoA) is another thiol compound that undergoes *S*-nitrosylation, and the resulting *S*-nitroso-CoA (CoA-SNO) mediates protein *S*-nitrosylation [158]. Interestingly, AKR1A catalyzes the NADPH-dependent reduction of CoA-SNO and also produces CoA-sulfinamide [4], which results in the suppression of protein *S*-nitration and the termination of NO signaling. Pyruvate kinase M2 is sensitive to S-nitosylation and, when inactivated, causes more glucose flow to the pentose phosphate pathway, resulting in an increase in NADPH production in *Akr1a*-KO mice [159]. As NADPH is a required electron donor for a variety of reductase activities, such as glutathione reductase and thioredoxin reductase, the antioxidant capacity of cells would also be elevated due to elevated *S*-nitosylation reactions, which may rationalize the resistance against thioacetamide [125] and the preservation of running performance [122] in Akr1a-KO mice. 

## 10. Concluding Remarks

While AKR1A plays pleiotropic functions that include ascorbate synthesis, aldehyde detoxification, and the metabolic conversion of bioactive compounds (Figure 8), the physiological functions of AKR1A are still not completely understood. Difficulty in specifying functions in vivo can partly be attributed to its broad substrate specificity and the presence of family members that exhibit similarities in substrate usage. There appear to be more functions attributable to AKR1A and these are being unveiled in mammalian physiology. The genetic modification of *Akr1a* could be useful for such a purpose by allowing AKR1A to be differentiated from other family enzymes. Moreover, the regulatory mechanism of the enzymatic activity for this enzyme is largely unknown and awaits clarification by further studies if a better understanding of the AKR1A functions is to be achieved.

## Figures and Tables

**Figure 1 metabolites-11-00343-f001:**
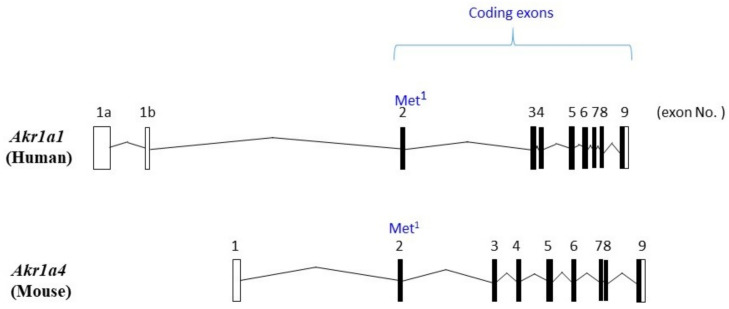
Structures of *Akr1a1* and *Akr1a4*. *Akr1a* is encoded by 10 and 9 exons in the human gene (*Akr1a1*) and the mouse gene (*Akr1a4*), respectively. The first 2 exons of *Akr1a1* encoding the 5′-untranslated region are alternatively transcribed [8], but both exons are occasionally transcribed together. Exons 2 to 9 encode protein sequences, so that alternative transcripts encode the same AKR1A protein sequences.

**Figure 2 metabolites-11-00343-f002:**
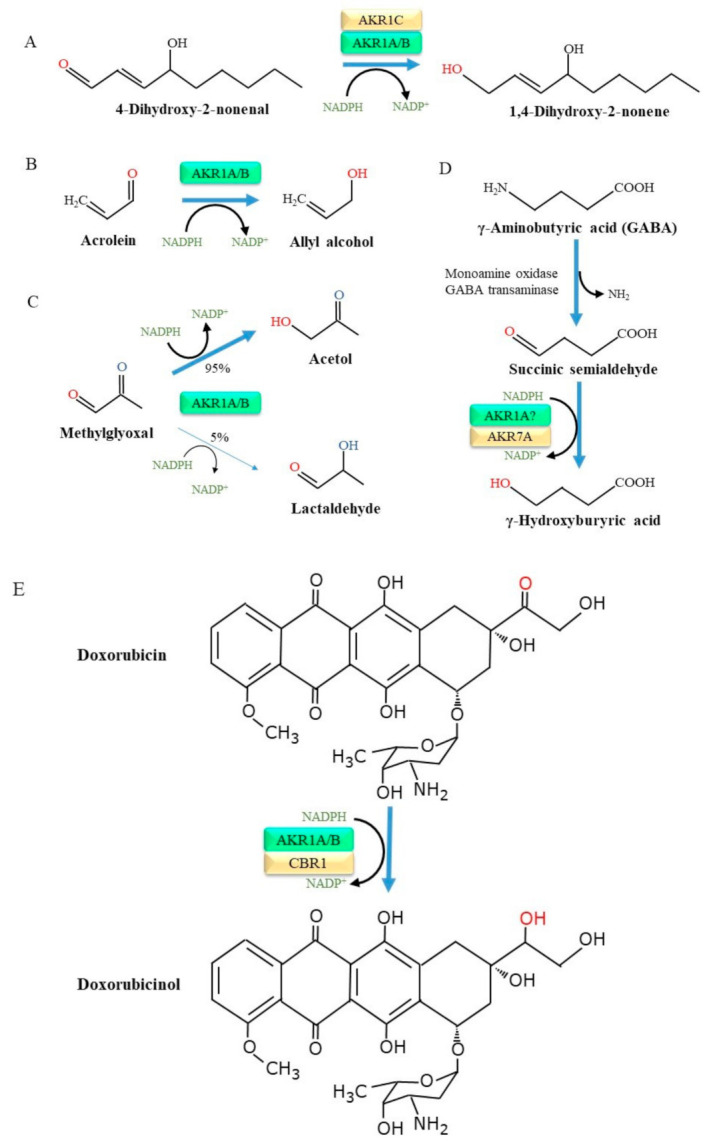
Examples for substrates and products of AKR1 enzymatic reactions. AKR enzymes catalyze the NADPH-dependent reduction of aldehyde molecules that are derived from lipids, carbohydrate, monoamines, and drugs. Shown are examples of the bioactive compounds; 4-dihydroxy-2-nonenal (**A**), acrolein (**B**), methylglyoxal (**C**), succinic semialdehyde (**D**), and doxorubicin (**E**).

**Figure 3 metabolites-11-00343-f003:**
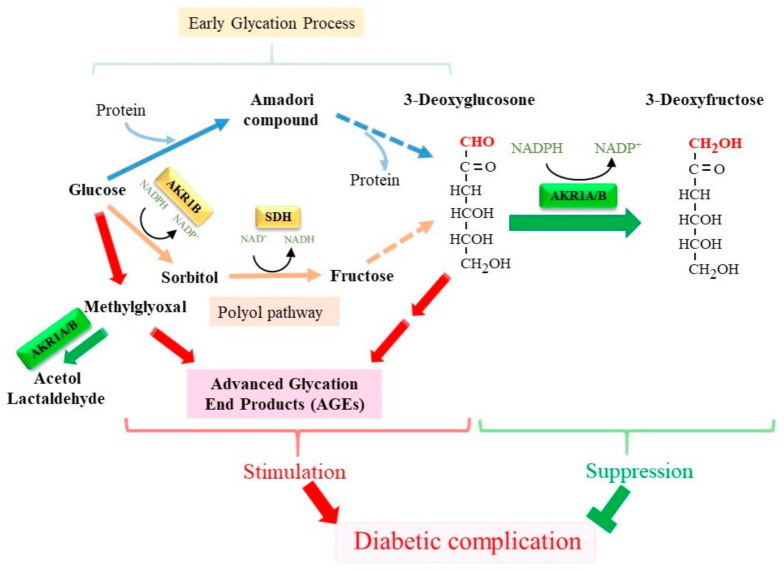
AKR1A/B may suppress diabetic complications by preventing the formation of AGEs. 3-DG is an intermediary compound that is non-enzymatically produced under hyperglycemia via an Amadori compound and advances to form AGEs, which results in the development of diabetic complications. AKR1A/B reduce 3-DG to 3-DF and consequently suppress the formation of AGEs. AKR1B catalyzes the first step of the polyol pathway by means of the conversion of glucose to sorbitol. Methylglyoxal, which is produced from glucose metabolism, is also involved in production of AGEs, while it is eliminated by glyoxalases or AKR1A/B as depicted in Figure 2C.

**Figure 4 metabolites-11-00343-f004:**
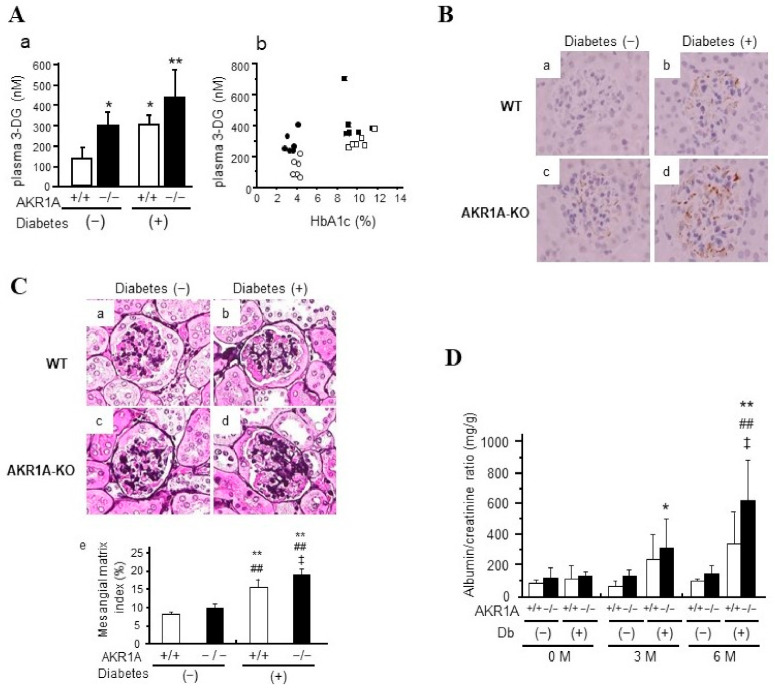
AKR1A suppresses diabetic nephropathy in mice. (**A**) Plasma 3-deoxyglucosone concentration. (**a**): Plasma 3-DG concentrations were measured in each mouse at 6 months after injection of STZ or vehicle. Data are the mean ± SD (n = 6 each). * *p* < 0.05, ** *p* < 0.01 vs. non-diabetic WT mice. (**b**): Relation between HbA1c and plasma 3-DG levels is plotted. White circles = non-diabetic WT mice; black circles = non-diabetic *Akr1a*-KO mice; white squares = diabetic WT; black squares = diabetic *Akr1a*-KO mice. (**B**) Pyrraline formation in renal glomeruli. The kidneys of WT and *Akr1a*-KO mice were removed after 6 months of STZ or vehicle treatment, and immunohistochemical studies using monoclonal mouse anti-pyrraline antibody were performed. (**a**): Non-diabetic WT mice; (**b**): diabetic WT mice; (**c**); non-diabetic *Akr1a*-KO mice; (**d**): diabetic *Akr1a*-KO mice. (**C**) Morphological changes in renal glomeruli. The kidneys of WT and *Akr1a*-KO mice were removed after 6 months of STZ or vehicle treatment, and PAM staining was performed. (**a**): Non-diabetic WT mice; (**b**): diabetic WT mice; (**c**); non-diabetic *Akr1a*-KO mice; (**d**): diabetic A *Akr1a*-KO mice. (**e**): The mesangial matrix index was calculated using the formula [(ECM area / glomerular tuft area) × 100 (%)]. Data are mean ± SD. ** *p* < 0.01 vs. non-diabetic WT mice; ## *p* < 0.01 vs. nondiabetic *Akr1a*-KO mice; ‡ *p* < 0.05 vs. diabetic WT mice. (**D**) Urine albumin/creatinine ratio (ACR). Urine ACRs of WT and *Akr1a*-KO mice were determined before and 3 months, and 6 months after STZ or vehicle treatment. * *p* < 0.05, ** *p* < 0.01 vs. non-diabetic WT mice; ## *p* < 0.01 vs. non-diabetic *Akr1a*-KO mice; ‡ *p* < 0.05 vs. diabetic WT mice.

**Figure 5 metabolites-11-00343-f005:**
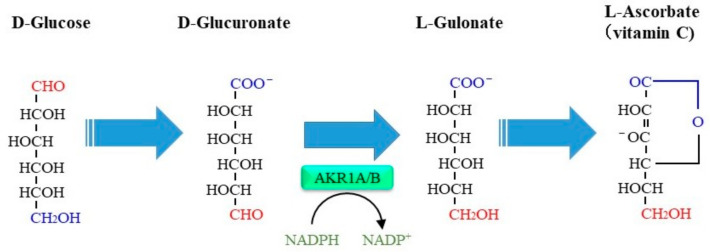
A role of AKR1A and AKR1B in ascorbate synthesis. AKR1A and AKR1B catalyze the reduction of d-glucuronate to l-gulonate which is a component of the ascorbate synthetic pathway in competent animals, but their contributions vary. The alternative, minor pathway of conversion of d-glucuronate to l-gulono-**γ**-lactone is not shown.

**Figure 6 metabolites-11-00343-f006:**
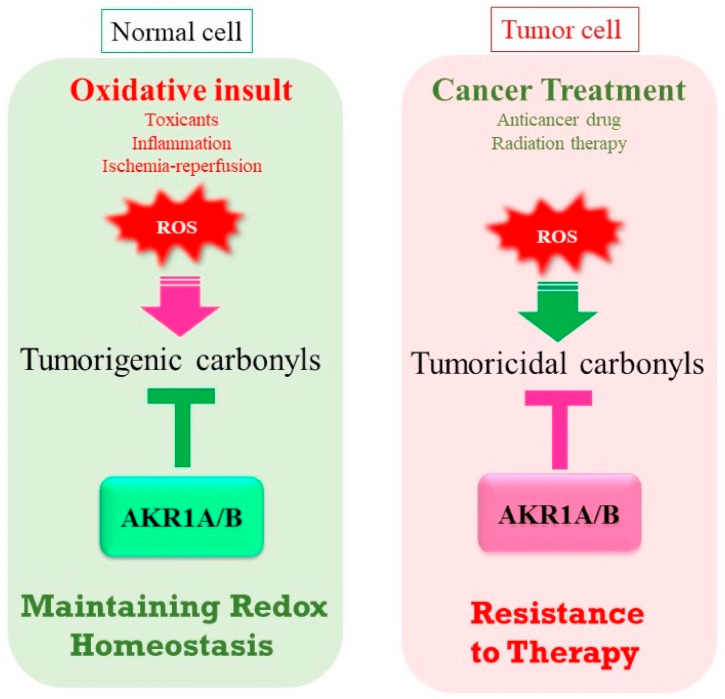
Antithetical roles of AKR1A/B in tumor development. In normal cells, AKR1A/B detoxify tumorigenic aldehyde compounds and hence exert anti-carcinogenic activity. On the other hand, tumor cells can become resistant to anti-cancer drugs and radiation therapy via the reduction of tumoricidal aldehydes.

**Figure 7 metabolites-11-00343-f007:**
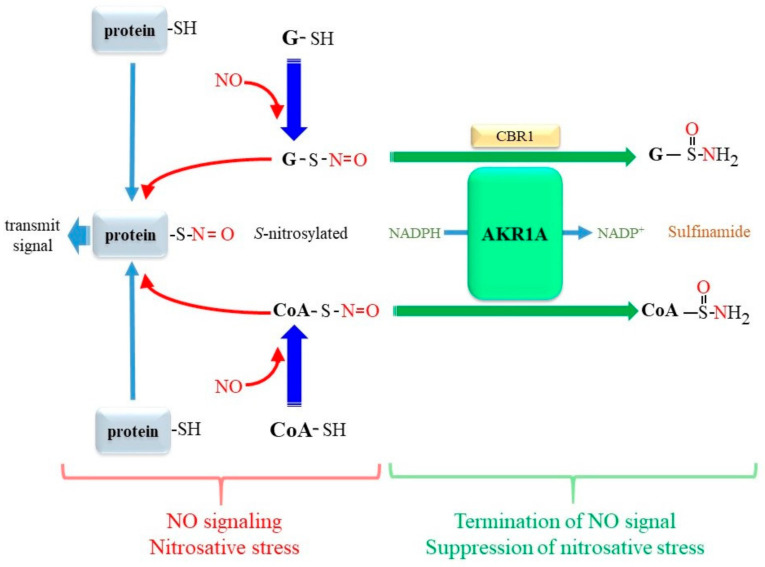
AKR1A reductively metabolizes GSNO and *S*-nitroso-coenzyme A. NO reacts with sulfhydryl groups of GSH and CoA and forms GSNO and *S*-nitroso CoA (CoA-SNO), which results in protein S-nitrosylation. AKR1A reductively converts GSNO and CoA-SNO to glutathione sulfonamide and CoA sulfonamide, respectively, and thereby suppresses NO signaling or nitrosative stress mediated by *S*-nitrosylated proteins.

**Figure 8 metabolites-11-00343-f008:**
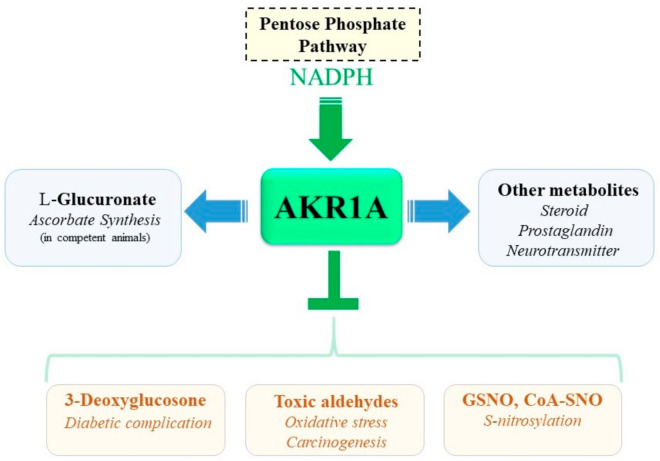
Multiple functions of AKR1A. AKR1A contributes to ascorbate synthesis in competent animals and also other metabolic pathways via NADPH-dependent reduction of aldehyde compounds. AKR1A also suppresses the formation of aldehyde intermediates, including the formation of 3-DG and lipid peroxidation products, leading to improved diabetic complications and oxidative damage. Nitrosative stress is consequently mitigated by means of the GSNO reductase and *S*-nitroso CoA reductase activity of AKR1A.

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
