# Peer review of "Pleiotropic Actions of Aldehyde Reductase (AKR1A)"

_metabolites, 2021, doi:10.3390/metabo11060343_

Round 1

Reviewer 1 Report

In this review article the authors describe the multifaceted effects of AKR1A, especially its relationship to AGE, its partial involvement in ascorbic acid biosynthesis, and its involvement in glutathione s-nitrosylation and coenzyme A reduction.  The physiological functions of AKR1A are systematically well described.  The research results of other groups as well as the research result of their own group are equally introduced. This review paper would be acceptable after minor correction.

1) In Fig.3, the green arrow of AKR1A / B appears from Methylglyoxal.  I think it would be better to write the name of this metabolized compounds, Acetol and Lactaldehyde.

2) The subtitles of section 5 and 6 are same (AKR1A as a suppressor of diabetic complications).  The content of 6 is about the biosynthesis of ascorbic acid, so there is a mismatch.  Please change the subtitle of section 6.

3) In this paper the authors use “D-glucuronate” and “L-gulonate” as a compound name.  But in p.11, line 383, the authors wrote “D-glucuronic acid” and “L-gulonic acid”.  Why?

Author Response

Thank you very much for kindly evaluating our manuscript. We have amended it according to your comments. Our responses follow your comments.

1) In Fig.3, the green arrow of AKR1A / B appears from Methylglyoxal.  I think it would be better to write the name of this metabolized compounds, Acetol and Lactaldehyde at the end of green arrow from Methylglyoxal.

Responses: Thank you for kind advice. We have written Acetol and Lactaldehyde.

2) The subtitles of section 5 and 6 are same (AKR1A as a suppressor of diabetic complications).  The content of 6 is about the biosynthesis of ascorbic acid, so there is a mismatch.  Please change the subtitle of section 6.

Responses: Thank you for kindly pointing out. According to the reviewer’s advice, we have corrected the subtitle of section 6 to “6. Roles of AKR1A in ascorbate biosynthesis and action in mice”.

3) In this paper the authors use “D-glucuronate” and “L-gulonate” as a compound name.  But in p.11, line 383, the authors wrote “D-glucuronic acid” and “L-gulonic acid”.  Why?

Responses: Thank you for kindly pointing out. We happened to use a mixture of terms. According to the reviewer’s advice, we have corrected the latter to the “D-glucuronate” and “L-gulonate” for consistency.

Reviewer 2 Report

Fujii et al. provide an overview of the physiological actions of the aldehyde reductase (AKR1A) in ascorbate synthesis, aldehyde detoxification, the synthesis of bioactive compounds, as well as important processes. The enzyme may also play important roles in diabetes and cancer. For example, the detoxification of aldehydes by AKR1A in a NADPH-dependent reaction may have anti-diabetic properties. Its reduction of intermediates involved in glycation can suppress formation of advanced glycation end products and thus ameliorate diabetic complications.

Specific Comments:

(1) In 'Section 3. Protein Structure and catalytic reaction of AKR1A', a structure of one of the AKR1 enzymes would be very useful to the reader. It would allow the authors to highlight active site residues, such as Lys-262 and Tyr-50, in the enzyme family. It could also provide some insight for the reader into the broad substrate utilization of the enzyme.

(2) In 'Section 4.4. Roles of the Akr1a in drug metabolism', the position of the glutamate and asparagine residues described in the second sentence of the second paragraph should be identified.

(3) Is the information shown in Figure 4 all unpublished?

Author Response

(The authors gave the same response as above.)

Reviewer 3 Report

The ms entitled „Pleiotropic actions of …” by Fuji et al. is a comprehensive review of the biology of AKR1A. The paper is very detailed and cites the most important observations.

Major point

Although the focus is according the title AKR1A, sometimes reference is made to AKR1B and C. For the non-expert reader it would be helpful if a table could be included that summarize the main differences of the different enzymes, including substrates, reaction type, cofactors, MW, expression sites, regulation etc.

Minor points:

Typos/formating error:

  1. 4 l. 162-167

p.6 l. 204 (HepG2)

Author Response

Thank you very much for kindly evaluating our manuscript. We have amended it according to your comments. Our responses follow your comments.

Major point

Although the focus is according the title AKR1A, sometimes reference is made to AKR1B and C. For the non-expert reader it would be helpful if a table could be included that summarize the main differences of the different enzymes, including substrates, reaction type, cofactors, MW, expression sites, regulation etc.

Responses: Thank you for kind advice. Unfortunately, enzymatic analyses on AKR1A in association with other family members have not been performed systematically. So there is only fragmentary information about AKR1A. This review article focuses on Akr1a and will be a part of a Special Issue on “Metabolic Regulation of Aldo-Keto Reductases”, so readers can find review articles on other genes overviewed by corresponding expert. In addition, substrates for AKR1A are too broad to cover, and there appears to be much more than we know. Accordingly, instead of summarizing the issue in Table, we occasionally mentioned it briefly in text and cited literatures that provide the information.  

Minor points:

Typos/formating error:

  1. 4 l. 162-167

p.6 l. 204 (HepG2)

Responses: Thank you for kindly pointing out. We have corrected them.